# Genome-Wide Identification and Characterization of Copper Chaperone for Superoxide Dismutase (CCS) Gene Family in Response to Abiotic Stress in Soybean

**DOI:** 10.3390/ijms24065154

**Published:** 2023-03-08

**Authors:** Shuang Jiao, Rui Feng, Yu He, Fengming Cao, Yue Zhao, Jingwen Zhou, Hong Zhai, Xi Bai

**Affiliations:** 1College of Life Science, Northeast Agricultural University, Harbin 150030, China; 2Key Laboratory of Soybean Molecular Design Breeding, Northeast Institute of Geography and Agroecology, Chinese Academy of Sciences, Harbin 150081, China

**Keywords:** *Glycine max*, *GmCCSs*, ROS, abiotic stress

## Abstract

Copper Chaperone For Superoxide Dismutase (CCS) genes encode copper chaperone for Superoxide dismutase (SOD) and dramatically affect the activity of SOD through regulating copper delivery from target to SOD. SOD is the effective component of the antioxidant defense system in plant cells to reduce oxidative damage by eliminating Reactive oxygen species (ROS), which are produced during abiotic stress. CCS might play an important role in abiotic stress to eliminate the damage caused by ROS, however, little is known about CCS in soybean in abiotic stress regulation. In this study, 31 *GmCCS* gene family members were identified from soybean genome. These genes were classified into 4 subfamilies in the phylogenetic tree. Characteristics of 31 *GmCCS* genes including gene structure, chromosomal location, collinearity, conserved domain, protein motif, cis-elements, and tissue expression profiling were systematically analyzed. RT-qPCR was used to analyze the expression of 31 *GmCCS* under abiotic stress, and the results showed that 5 *GmCCS* genes(*GmCCS5*, *GmCCS7*, *GmCCS8*, *GmCCS11* and *GmCCS24*) were significantly induced by some kind of abiotic stress. The functions of these *GmCCS* genes in abiotic stress were tested using yeast expression system and soybean hairy roots. The results showed that *GmCCS7*/*GmCCS24* participated in drought stress regulation. Soybean hairy roots expressing *GmCCS7*/*GmCCS24* showed improved drought stress tolerance, with increased SOD and other antioxidant enzyme activities. The results of this study provide reference value in-depth study CCS gene family, and important gene resources for the genetic improvement of soybean drought stress tolerance.

## 1. Introduction

Abiotic stress, such as drought, high salinity, low and high temperature, heavy metals can seriously affect the normal growth and development of plants, especially the yield of crops. Drought stress can reduce the productivity of soybean by interfering with physiological and biochemical processes such as photosynthesis, translocation, respiration, and growth stimulants [1,2]. During a prolonged drought stress condition, reaction oxygen species (ROS) accumulate excessively and cause oxidative damage. Reaction oxygen species (ROS) are produced by redox reactions in plants, including aerobic respiration and photosynthesis, and modulate some important pathways, including apoptosis, energy metabolism, the inflammatory response, and stress responses. High levels of ROS such as O_2_ (singlet oxygen), $O_{2}^{\bar{.}}$ (superoxide radical), OH (hydroxyl radical) and H_2_O_2_ (hydrogen peroxide) are generated during drought stress. To overcome drought-induced damage, plants activate their defense system by enhancing superoxide dismutase (SOD) activity to scavenge superfluous ROS. 

CCS (copper chaperone for copper-zinc superoxide dismutase) proteins work as a copper chaperone and this factor has been characterized as a specific chaperone for SOD (superoxide dismutase). CCS proteins were identified in the human first time which transferred, interfere and balanced copper to SOD and plan a strategy for reducing or preventing amyotrophic lateral sclerosis (ALS) [3,4]. SODs are metalloenzymes and the active site can be classified into Cu, Zn-SOD, Mn-SOD, or Fe-SOD depending on the metal. The CCS would recruit the copper ions to the copper-requiring enzyme Cu-zine (Zn) superoxide dismutase (SOD1) whereas increasing the portion of CCS remarkably extends the survival of the transgenic mice administered with CuII (astm) [5]. A CCS-null mutant of *Drosophila* had characterized the requirement for CCS in activating SOD1, and was founded that the resemblant phenotypically of shortened adulthood [6]. The chloroplastic copper chaperone for the cuprozinc superoxide dismutase *GmCCS* gene was demonstrated to exist and cloned in 2014 in soybean [7]. A human liver fluke, Clonorchis sinensis, CsSOD1 can be successfully activated by the full-length CsCCS, while deletion of the N-terminal domain of CsCCS did not function [8]. CCS selectively bound to and facilitated Cu transfer to MEK1 in turn modulates the canonical mitogen-activated protein kinase pathway consisting of the RAF-MEK-ERK signaling cascade [9,10].

Nevertheless, the biological functions and molecular mechanisms of the *GmCCS* have not been revealed in soybean. The *GmCCS* members were identified in this study, and their location in the genome was determined. The motif, gene structure, conserved domains, and transcription factor prediction were also analyzed. The expression patterns of these genes in different tissues and under different stress conditions were analyzed. A series of abiotic stress models in yeast systems and hairy roots of soybean were tested. The mechanism of how this gene regulates oxidase activity during the biological stress response is gradually revealed. This study provides insights into the *GmCCS* gene family in *Glycine (G.) max* and a theoretical basis for future research on *GmCCSs*.

## 2. Results

### 2.1. Genome-Wide Identification of the GmCCSs Family of Glycine (G.) Max Subsection

To identify putative *GmCCS* genes, sequences from 3 *Arabidopsis* CCS proteins were used as queries in BLASTp search against *Glycine (G.) max* with relatively complete genome annotations. 31 *GmCCS* family members were identified and named *GmCCS1*-*GmCCS31* in soybean. The CDS length of *GmCCSs* ranged from 246 bp to 960 bp, the protein length was between 82 aa and 320 aa and PIs of 4.65–9.41. Nearly 70% of *GmCCSs* PI is less than 7, which is inferred to be an acidic protein (Appendix A). Subcellular location prediction showed that 17 *GmCCS* members were located in the nucleus, 11 *GmCCS* members were located in the golgi apparatus and 5 *GmCCS* members were located in the cell membrane. Among them, 3 *GmCCS* members(*GmCCS1*, *GmCCS11*, *GmCCS30*) were located in the golgi apparatus and nucleus. *GmCCS13* was located in the cell membrane and chloroplast. 

Chromosomal location indicated that 31 *GmCCS* genes excited on all ten chromosomes (Chrs). Thirty-one putative *GmCCS* were located on Chr19, followed by *Chr11*(5), *Chr03*(4), *Chr12*(4), *Chr05*(3), *Chr04*(2), *Chr10*(2), *Chr16*(2), *Chr20*(2), *Chr13*(1) (Figure 1A). 

The orthologous relationship between *GmCCS* genes in *Glycine (G.) max*, *Arabidopsis* and *Glycine (G.) soja* genomes was assessed to clarify the evolutionary differences between *CCS* genes. *CCSs* were collinear with 47 and 15 genes in *Glycine (G.) soja* and *Arabidopsis*, respectively. The results showed that the *Glycine (G.) max* was more closely related to legume crops than other plants (Figure 1B). Gene duplication events are common in all species and generate new functional genes and drive evolution, 17 pairs of *GmCCSs* in *Glycine (G.) max* had collinearity among all 20 chromosomes (Figure 1C).

The maximum likelihood method was used to perform the phylogenetic analysis of GmCCS members of soybean (*Glycine (G.) max*), *Arabidopsis thaliana* and *Saccharomyces cerevisiae*. The model for MEGA prediction was used to analyze the phylogenetic relationship of *GmCCS* family. The *GmCCS* family could be classified into 4 subgroups (Groups I, II, III, IV) based on the phylogenetic tree analysis (Figure 2). GmCCSs have the same branch as different AtCCSs and ScCCS, which indicates that compared with other species, *Glycine (G.) max* is closed related to *Arabidopsis thaliana* and *Saccharomyces cerevisiae*. The yeast ScCCS1 is a multifunctional chaperone promoting all levels of SOD1 maturation [11].

### 2.2. Gene Structure and Conserved Domain Analysis

Genome sequence and annotation files of AtCCSs, ScCCS and GmCCSs were downloaded and queried by TBtools. The exon-intron organization of 31 *GmCCS* genes was examined to obtain information on the structure, diversity, and evolution of GmCCS families in soybean, and the number of exons in GmCCS family members were from 3–7 (Figure 3B). Closely related genes have similar exon-intron arrangements with similar protein structures.

In addition to the exon-intron pattern, the conserved domain could also be major for the various functions of *GmCCS* genes. Sequence analysis showed that the GmCCS proteins in soybean could be grouped as distinct clusters, more than half of all GmCCS proteins have the Cu-zine (Zn) superoxide dismutase conserved domain (Figure 3A). Previous study showed that the CCS would recruit the copper ions to the copper-requiring enzyme Cu-zine (Zn) superoxide dismutase (SOD1) whereas increasing the portion of CCS remarkably extends the survival of the transgenic mice administered with Cu^II^ (astm) [5]. It is worth noting that the twelve members of GmCCS proteins have heavy-metal-associated (HMA) domain, which directly binds AVR-Pik to activate plant defense during the stress of adversity [12]. 

### 2.3. The Motif, Cis-Regulatory Elements and Transcription Factor Analysis of GmCCSs

The MEME software was used to analyze the motifs of GmCCSs in *Glycine (G.) max*. Five different colored squares on the line represent different motifs in *GmCCS* members. The position on line indicates the composition of each CCS family member. Although various subgroups have different motifs, the *GmCCSs* members in the same subgroup have similar motif structures. For example, motif 4 is founded in subgroup I and subgroup IV (Figure 4A).

The analysis of promoter cis-elements can provide ideas for the tissue-specific expression and stress response modes of genes. The cis-regulatory elements of the 2000 bp promoter sequence of *GmCCS* genes were analyzed using PlantCARE. Twenty-two cis-regulatory elements were identified in the *GmCCSs* promoter sequence, including hormone-, stress- and photoresponsive element (Figure 4B). Light-responsive related elements were found in subgroup I and subgroup III, including TCT-motif, G-Box, GA-motif and GAGA-motif, TGA-box related to auxin-responsive [13]. Almost all members of GmCCS include stress-responsive elements, for instance, ABRE which response to ABA [14,15].

Promoter cis-elements combine with TFs to regulate the precise initiation and efficiency of transcription. Therefore, potential TFs that bind to the *GmCCSs* promoter were predicted by PlantTFDB (Figure 5). Almost all members were involved in the regulation of MADS, which responds to control plant flower development process [16,17,18]. DOF is involved to specific gene expression of stomatal guard cells and plant stress defense [19,20]. For instance, ERF, MYB, NAC, bZIP and WRKY transcription factors which correspond to biological stress and abiotic stress were found in 24 GmCCS members. ERF proteins were confirmed to control cell cycle and DNA repair by regulating genes. The PheE2F-PheDP complex exhibited diverse expression patterns in response to drought and salt treatment and diurnal cycles [21]. bZIP transcription factor regulates their target genes on their promoters and thereby direct adjusts plant developmental and abiotic stresses, such as drought, salinity, and cold. S1-bZIP is a unique one, which conserved upstream open reading frames mediate Sucrose-Induced Repression of Translation(SIRT), maintain sucrose homeostasis and withstand abiotic stresses in plants [22]. LcMYB2 promoted seed germination and root growth under drought and ABA treatments. LcMYB2 was confirmed in sheepgrass can regulate LcDREB2 expression by binding the promoter of LcDREB2, thereby, activating the expression of the osmotic stress marker genes in transgenic *Arabidopsis* [23].

### 2.4. Expression Analysis of GmCCS Genes in Different Tissues

To better understand the function of *GmCCS* genes in soybean (*Glycine (G.) max*), the expression profiles of *GmCCS* genes in different tissue, root, stem, leaf, flower and pod e.g., were analyzed using transcriptome data from the Phytozome (Figure 6A). The hierarchical clustering of the *GmCCS* genes was generated using a heatmap with the average fragments per kilobase of transcript per million mapped reads (FPKM) values in triplicate (Figure 6). 14 *GmCCS* genes exhibited high transcript abundance(FPKM > 5) in the root, followed by 11 *GmCCS* genes in the leaves, and 12 *GmCCS* genes in the hypocotyl, the expression of *GmCCS9* was not detected in any of the analyzed tissues. Overall, the expression levels of 8 *GmCCS* genes from I and II subfamilies accumulated (FPKM > 5) in all of the tested tissues, which suggests that these genes play a crucial role in tissue development.

### 2.5. Expression Profiles of the Candidate GmCCS Genes under Different Abiotic Stresses

RT-qPCR was performed to analyze profiles of *GmCCS* genes under different abiotic stresses. The heatmap was created based on the RT-qPCR results. As shown in Figure 7A, transcript levels of *GmCCS7*, *GmCCS8* and *GmCCS24* were greatly induced by drought stress at some time points in both leaf and root. *GmCCS11* was greatly induced by drought stress at some time points only in leaf and *GmCCS5* was induced by drought stress at some time points only in root. These five *GmCCS* genes all respond to drought stress at transcript levels. The transcript level of the other *GmCCS* genes did not show significantly difference in drought stress during a series of treatment times. The transcript level of *GmCCS5* was greatly induced by NaCl stress at 2 h in leaf (Appendix A). The transcript level of the other *GmCCS* genes did not show significantly difference upon NaCl stress during a series of treatment times (The fold change of gene expression was less than 1 fold.). Under NaHCO_3_ stress treatment, transcript levels of *GmCCS14*, *GmCCS10*, *GmCCS22*, *GmCCS5* and *GmCCS6* were induced at 1 h in leaf (Appendix A). However the fold changes of these five genes’ expression were less than 1 fold. Above results indicated that *GmCCS5*, *GmCCS7*, *GmCCS8*, *GmCCS11* and *GmCCS24* might be involved in drought stress regulation, and *GmCCS5* might be involved in drought and salt stress regulation.

### 2.6. Overexpression of Abiotic Candidate GmCCS Genes Improved Drought Tolerance in Yeast Cells

To analyze whether these five abiotic responses *GmCCS* genes are involved in abiotic stress tolerance, *GmCCS5*, *GmCCS7*, *GmCCS8*, *GmCCS11* and *GmCCS24* were subcloned into pYES2 vector, and overexpressed in yeast cells respectively. Yeast cells transformed with an empty pYES2 vector was as the control. Spot assay grown on medium supplemented with sorbitol, NaCl, and NaHCO_3_ was performed. Using sorbitol simulating drought stress, when the concentration of supplemented sorbitol was 0.4 M, expression of these five *GmCCS* genes dramatically enhanced sorbitol tolerance of yeast cells compared with control yeast cells (Figure 8). When sorbitol was supplemented with 0.8 and 1.6 M, expression of *GmCCS5*, *GmCCS7*, *GmCCS8* and *GmCCS11* dramatically enhanced sorbitol tolerance of yeast cells, however, transformants expressing *GmCCS5* did not (Figure 8). When sorbitol was supplemented with 2.0 M, transformants expressing these five *GmCCS* genes or the empty vector did not grow on the medium, indicating the concentration of sorbitol was too high. Under NaCl and NaHCO_3_ stress, expression of these five genes did not enhance NaCl or NaHCO_3_ tolerance of yeast cells (Appendix A). These five *GmCCS* genes all enhanced the drought tolerance of yeast, and *GmCCS7*, *GmCCS8*, *GmCCS11* and *GmCCS24* showed more stronger drought resistance.

### 2.7. GmCCS7 and GmCCS24 Improve Drought Stress Tolerance in Transgenic Soybean Hairy Roots

To further confirm the drought stress tolerance of these *GmCCS* genes, a soybean transgenic hairy root system was used. Since *GmCCS7*, *GmCCS8*, *GmCCS11* and *GmCCS24* showed stronger drought resistance in yeast cells, we intended to further test the drought stress tolerance of these four genes. The amino acid identities of *GmCCS7* and *GmCCS8* were 97.37% and *GmCCS7* encoding protein had 8 more amino acids than *GmCCS8*, raising the possibility that *GmCCS7* represents the function of *GmCCS8.* Thus, *GmCCS7*, *GmCCS11* and *GmCCS24* were overexpressed in soybean hairy roots for drought stress tolerance analysis. 

Accordingly, we overexpressed three *GmCCSs* under the control of the CaMV 35S promoter in the soybean hair roots. Compared to control hairy roots, transgenic roots had a significant increase of *GmCCS7*, *GmCCS11* and *GmCCS24* expression, respectively (Figure 9A). Soybean seeding plants with *GmCCS7-OE* and *GmCCS24-OE* hair roots performed better than the control, however *GmCCS11-OE* did not (Figure 9B). Survival rates (%) were calculated after drought stress treatment. As shown in Figure 9C, survival rates were 67.3% and 73.2% for soybean seeding plants with *GmCCS7-OE* and *GmCCS24-OE* root, 13.9% for seeding plants with *GmCCS11-OE*, and 27.3% for control plants. The survival rates of soybean seeding plants with *GmCCS7-OE* and *GmCCS24-OE* root were significantly higher than the control plants, however, survival rates of soybean seeding plants with *GmCCS11-OE* root were not. These results indicated that *GmCCS7* and *GmCCS24* improve drought stress tolerance in soybean, but *GmCCS11* does not.

To explore the mechanism of *GmCCS7*, *GmCCS11* and *GmCCS24* in drought stress tolerance, the activities of antioxidant enzymes (SOD, CAT, POD) were examined. As shown in Figure 9D–F, the SOD, CAT, and POD activities of three *GmCCS-OE* transgenic soybean roots showed no differences compared with control plants without drought treatment. Under drought stress conditions, the activities of these three enzymes in *GmCCS7-OE and GmCCS24-OE* transgenic soybean roots were significantly higher than those in control plants. The activities of CAT in *GmCCS11-OE* transgenic soybean roots were higher than those in control soybean roots, however, the activities of SOD and POD in *GmCCS11-OE* transgenic soybean roots were comparable to the control plants. *GmCCS11* could enhance the activities of antioxidant enzymes CAT, but not SOD and POD, raising the possibility of why *GmCCS11* did not improve drought stress tolerance in soybean. MDA content which represents the degree of cell membrane damage was assessed, as shown in Figure 9G, MDA contents of *GmCCS7-OE* and *GmCCS24-OE* transgenic soybean roots were lower than the control plants. Thus, the degree of damage to the cell membrane damage of *GmCCS7-OE* and *GmCCS24-OE* transgenic soybean roots was lower than the control. These results indicated that *GmCCS7* and *GmCCS24* enhanced drought stress tolerance by elevating the activities of antioxidant enzymes and decreasing cell membrane damage.

## 3. Discussion

Drought is an important factor that seriously affects the production and quality of soybean, especially for China which possesses a population base. The copper chaperone for superoxide dismutase (CCS) has been identified as a key factor in the integration of copper/zine superoxide dismutase (Cu/Zn SOD) in yeast and mammals [24,25]. In Arabidopsis (*Arabidopsis thaliana*), *AtCCS* has been proven as the activation of Cu/Zn SOD activity [26]. However, studies to identify the biological function of *GmCCS* genes family under abiotic stress were little. In our study, a total of 31 GmCCS family genes were identified in this study in the soybean by comparative analysis with the *Arabidopsis* CCS gene family. The GmCCSs protein containing heavy-metal-associated (HMA) domain and copper chaperone for copper-zinc superoxide dismutase conserved domain, these two conserved domains were necessary for *GmCCS* genes in different plants to perform biological function. Rice HMA plant protein 19, which directly binds AVR-Pik to activate plant defense during the stress of adversity [12]. Cu chaperone CCS achieves appropriate installation of Cu within the SOD and activation of SOD, which in turn mediates degeneration in the fetal motor neuron disorder amyotrophic lateral sclerosis. *AtCCS* gene was responsive to the activation of all three types of CuZnSOD activity in *Arabidopsis*. To date, very little is known about the functional development of copper chaperones in soybean, in our study, the essential information about GmCCS family members were systematically and specifically elaborated.

We identified 31 *GmCCS* family members, were through two rounds of BLASTP based on 3 *Arabidopsis* CCS. Gene duplication is the most common evaluation mechanism for gene family expansion, which plays an important role in evolution by facilitating the generation of new genes and gene functions. There are three main evolutionary modes of gene duplication: segmental duplication (fragment duplications), tandem duplication, and translocation events. Tandem and fragment duplications are the major gene replication events in plant gene family expansion. Herein, several gene clusters were found on all 17 chromosomes of the common bean, which is probably due to several gene duplication events (Figure 1). CCS protein plays a key role in various cellular processes, participates in plant growth, development and stress responses [9]. The study on the function of CCS protein will help to elucidate the mechanism by that plants respond to environmental stresses and provide the theoretical basis for environmental adaptation in plants. Recently, *MaCCS* proteins have been found in Musa acuminata cv. *Tianbaojiao*, is involved in Abiotic and Hormonal Stress Responses [27]. 31 *GmCCS* genes in soybean and classified into four subfamilies, AtCCS2 and ScCCS were distributed in Groups I, which contains 9 GmCCS family members. It implies that there are similar functions and more closely related in the evolution of *CCS* gene (Figure 2). More than half of all GmCCS proteins have the Cu-zine (Zn) superoxide dismutase conserved domain. And twelve members of GmCCS proteins have heavy-metal-associated (HMA) domain, which directly binds AVR-Pik to activate plant defense (Figure 3). 

The genes in each subgroup had similar motifs, gene structures, and cis-regulatory elements, supporting the reliability of the subfamily classification. The arrangement and number of exons and introns can provide important information about evolutionary relationships. Exon number plays a vital role in gene family evolution, while introns regulate gene expression. Therefore, gene function can be elucidated by analyzing gene structure. Generally, closely related genes have similar exon-intron arrangements with similar protein structures. In this study, the CCS genes in the same subgroup have the same number of exons and introns, indicating that they may have similar functions (Figure 3). This difference in the number of exons and introns of different genes in the same subfamily may be due to the functional diversity of genes during evolution.

Analysis of the arrangement and number of protein motifs in each CCS further demonstrated the categorization of the CCS family. The results showed that the structures were highly similar within each subfamily but different in each conserved region of the 5 motifs. Promoter analysis of CCS genes showed that CCS associated with stress (MBS, ARE, MRE and TC-rich), the light response (TCT-motif, G-Box, GA-motif and GAGA-motif, TGA-box) and hormone-related (ABR) genes broadly existed (Figure 4B). Promoter cis-elements combine with TFs to regulate the precise initiation and efficiency of transcription. ERF, MYB, NAC, bZIP and WRKY transcription factors which correspond to biological stress and abiotic stress were found in 24 *GmCCS* members(Figure 5). These results indicated that *GmCCS* genes may also be involved in various stresses and the regulation of different biological processes.

*GmCCS5*, *GmCCS7*, *GmCCS8*, *GmCCS11* and *GmCCS24* were significantly expressed in leaf and root in the present study and these genes are important regulators in the response to drought stress (Figure 6 and Figure 7). These GmCCS genes were distributed in the Group I through phylogenetic analysis (Figure 2), and these genes contain heavy-metal-associated (HMA) domain and copper chaperone for copper-zinc superoxide dismutase conserved domain (Figure 3). Moreover, the plants which transformed with *GmCCS7-OE* and *GmCCS24-OE* showed a lesser wilting phenotype than the negative control plants under drought stress, *GmCCS11-OE* can not withstand drought stress (Figure 9A). On the basis of these collective reasons, transgenic soybean overexpressing *GmCCS7* and *GmCCS24* exhibited an increase in antioxidant enzymes under drought tolerance.

## 4. Materials and Methods

### 4.1. Identification of the GmCCS Gene Family

The whole genome sequence and annotation files of *Glycine (G.) max* were downloaded from the laboratory Wm82.a2.v1 (Phytozome; https://phytozome-next.jgi.doe.gov/) (accessed on 24 March 2022). The details of genome annotations were uploaded as Appendix A, including gene sequences, CDS, protein sequences and gff3 files. AtCCSs protein sequences were downloaded from TAIR (TAIR; http//www.arabidopsis.org). BLAST searches using TBtools (Auto Blast Two Sequences Set E-value = 1 × 10^−5^) were performed using AtCCSs protein sequences as query sequences (Chen et al., 2020). Putative *GmCCSs* obtained by BLAST searches were then confirmed by NCBI-CDD and Pfam searches for the presence of the heavy-metal-associated domain (IPR006122) or copper chaperone for Superoxide dismutase domain (IPR024134). ExPASy-proparam was used to predict the coding sequence (CDS) length, predicted isoelectric (pI) and molecular weights (MWs) of all *GmCCSs*, Plant-mPLo 2.0 was used to predict their subcellular localization.

### 4.2. Gene Structure, Motif and Conserved Domain Analysis

The conserved motifs of the genes were analyzed by the MEME program with the following parameters: optimal motif width was set to 30–70, the number of repetitions was set to zero or one, the maximum number of motifs was set to identify 10 motifs, and visualized using TBtools. Gene structure and functional domains were analyzed and visualized using NCBI Batch CD-Search and TBtools.

### 4.3. Phylogenetic, Cis-Acting Element, Chromosomal Location Analysis, and Interacting Protein Prediction

CCS family protein sequences (Appendix A) of *Arabidposis*, *Glycine (G.) max*, *Saccharomyces cerevisiae* were downloaded from Phytozome database. A maximum likehood (ML) phylogenetic tree was constructed using MEGA 7.0, with the following parameters: Poisson correction, pairwise deletion, and bootstrap (1000 replicates). The 2000 bp promoter fragments upstream of start codon were obtained using TBtools [28]. The *cis*-acting element searches were performed using the putative *cis*-element databases PlantCare (http://bioinformatics.psb.ugent.be/webtools/plantcare/html/) and visualized using TBtools after the statistical screening. The chromosomal locations of *GmCCSs* were obtained from the NCBI database and visualized using TBtools. The interacting protein with GmCCSs was performed using PlantTFDB (http://planttfdb.cbi.pku.edu.cn/). 

### 4.4. Tissue Expression Profiling Analysis of GmCCS Gene Family in Soybean

Transcription data were obtained from NCBI to analyze the tissue expression patterns of *GmCCS* gene family. Gene expression level was calculated by FPKM values and illustrated using TBtools. 

### 4.5. Plant Materials and Treatments

Soybean cultivar Williams 82 was used to analyze the expression patterns of *GmCCS* genes under abiotic stresses. Seedings were grown in pots in the greenhouse with 16 h-light/8 h-dark photoperiod, 28/20 °C day/night temperatures, and 60% relative humidity. Salt/NaHCO_3_ stress treatment: soybean seeding was irrigated with 300 mM NaCl and 75 mM NaHCO_3_. The leaves and roots of seedlings were sampled at 0 h, 0.5 h, 1 h, 2 h, 4 h, 6 h, 9 h and 12 h time points during treatments. Drought stress treatment: Watering was halted. The moisture of soil was monitored. When the moisture of soil decreased to 30%, the leaves and roots of seedlings were sampled at 0 h, 0.5 h, 1 h, 2 h, 4 h, 6 h, 9 h and 12 h time points during drought treatments.

### 4.6. RNA Extraction and RNA Extraction and Reverse Transcription-Quantitative PCR (RT-qPCR) Analysis 

Total RNA was extracted from soybean leaves and roots of seedlings using the manufacturer’s protocol (Qiagen, Hilden, Germany). The isolated RNA was then subjected to reverse transcription by using the SuperScript^TM^ III Reverse Transcriptase kit. RT-qPCR was performed on each cDNA sample with the SYBR Green Master Mix on an ABI 7500 sequence detection system (Applied Biosystems, Waltham, Massachusetts, MA, USA). The measured Ct values were converted to relative copy-numbers using the ∆∆Ct method. Relative fold enrichment was calculated by normalizing the amount of a target DNA fragment against a reference gene *TUA5*. Three fully independent biological replicates (n = 3 plants) were obtained and subjected to RT-qPCR in technical triplicates. Raw data were standardized as described previously [29]. 

### 4.7. Stress Tolerance Analysis in Yeast

The coding sequence of *GmCCSs* was PCR amplified using cDNA from Williams 82 leaves with the primer pair listed in Appendix A. The PCR products were then inserted in the pYES2 vector. The resulting construct was then introduced into yeast strain INVSc1. Stress tolerance test in yeast was performed according to Gautam [30]. Yeast cells containing the vectors pYES2 (control) or pYES2-GmCCSs were grown in SC-URA (without uracil) medium with shaking (200 rpm) at 28 °C for 24 h, and their OD600 was adjusted with SC-URA to 1, 0.1, 0.01, 0.001, respectively. For the salt stress test, one microliter of each diluted culture was spotted onto SC-URA medium supplemented with 0.8, 1.0, 1.1, 1.3 M NaCl; For the osmotic stress test, the yeast cells were cultured in 0.4, 0.8, 1.6, 2.0 M sorbitol. For the alkali stress test, the yeast cells were cultured on SC-URA medium supplemented with 0.2, 0.4, 0.6, 0.8 M NaHCO_3_ at 28 °C. Transgenic yeast harboring empty vector pYES2 was used as the control. The experiments were repeated three times.

### 4.8. Stress Tolerance Analysis in Transgenic Soybean Hairy Toots 

The full-length *GmCCS7*/*GmCCS11*/*GmCCS24* coding sequence was inserted downstream of the CaMV 35S promoter and fused to the GFP, in the pCAMBIA1300 vector respectively. The resulting construct was then introduced into *A. rhizogenes* strain K599. To generate transformed soybean hairy roots, the soybean cultivar Dongnong50 (DN50) was used for *A. rhizogenes*-mediated transformation [31]. Seeds were germinated under a 16 h light/8 h dark photoperiod at 25 °C in a humidity chamber. After a week, healthy plants were injected with *A. rhizogenes* strain K599 harboring pCAMBIA1300 (CK) or pCAMBIA1300-GmCCSs construct vectors. The infected plants were then transferred to the chamber and kept under high humidity until hairy roots grew out of the infection sites to 2–5 cm in length [32]. The original roots were removed before subjecting to stress treatment, then the seedings with 2–5 cm hairy roots were transferred into Hoagland solution with 20% (*w*/*v*) PEG 6000 for 6 h drought stress treatment. Then the plants were transferred to fresh medium without PEG 6000. Survival rates (%) were calculated from the number of surviving plants per total plants tested. Three independent experiments were conducted with individual samples containing 15 plants. Antioxidant enzyme (SOD, CAT, POD) activity measurement was performed as described previously [33].

## 5. Conclusions

In this study, 31 *GmCCS* genes were identified from the soybean. The *GmCCS* members were divided into four subgroups based on the analysis of the composition, phylogenetic tree, gene structure, motifs, cis-regulatory elements, and collinearity. Members in each subgroup had similar analysis results. Transcriptome data analysis showed that *GmCCS* genes were specifically expressed in different tissues. RT-qPCR showed that five *GmCCS* genes responsed to drought stresses at transcript levels. Overexpression of *GmCCS7* and *GmCCS24* increased drought tolerance in soybean hair root and yeast. 

## Figures and Tables

**Figure 1 ijms-24-05154-f001:**
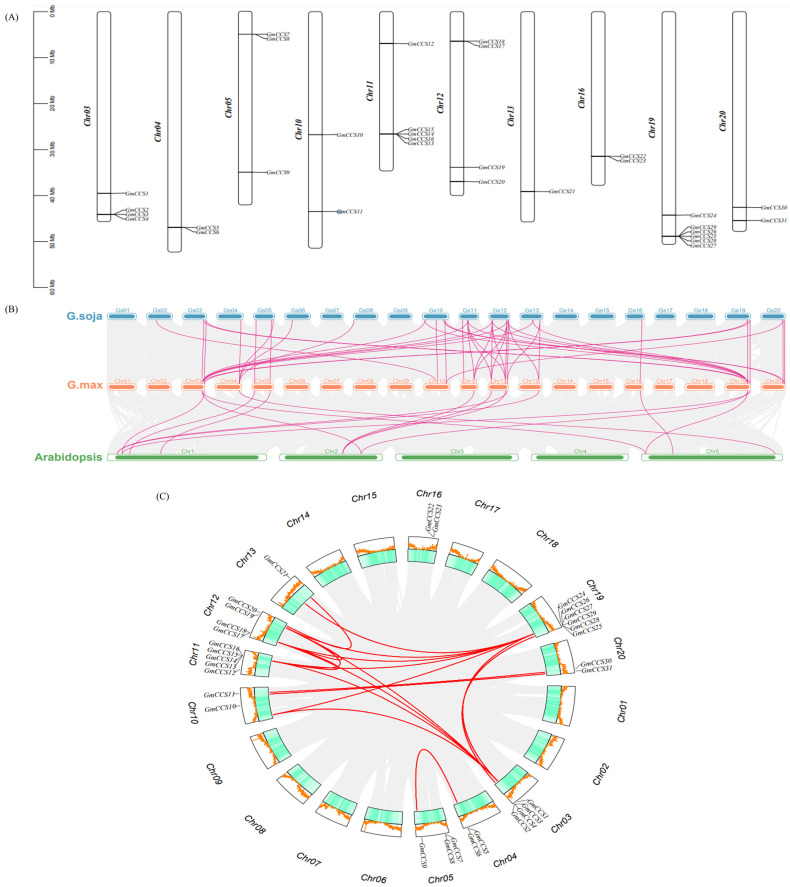
Chromosomal analysis of *GmCCS* members. (**A**) Chromosomal location of *GmCCSs* gene family in *Glycine (G.) max*. (**B**) Collinear analysis of GmCCSs with *Arabidopsis* and *Glycine (G.) soja*. Green, orange and blue represents *Arabidopsis*, *Glycine (G.) max* and *Glycine (G.) soja*, respectively. The bright lines indicate the connection with *GmCCSs*. (**C**) Chromosomal locations of *GmCCS* genes on 20 chromosomes of *Glycine (G.) max* and synteny analysis of interchromosomal relationships of *GmCCS* genes. The green blocks indicate parts of the radish chromosomes. The red lines indicate segment duplicated *GmCCS* gene pairs.

**Figure 2 ijms-24-05154-f002:**
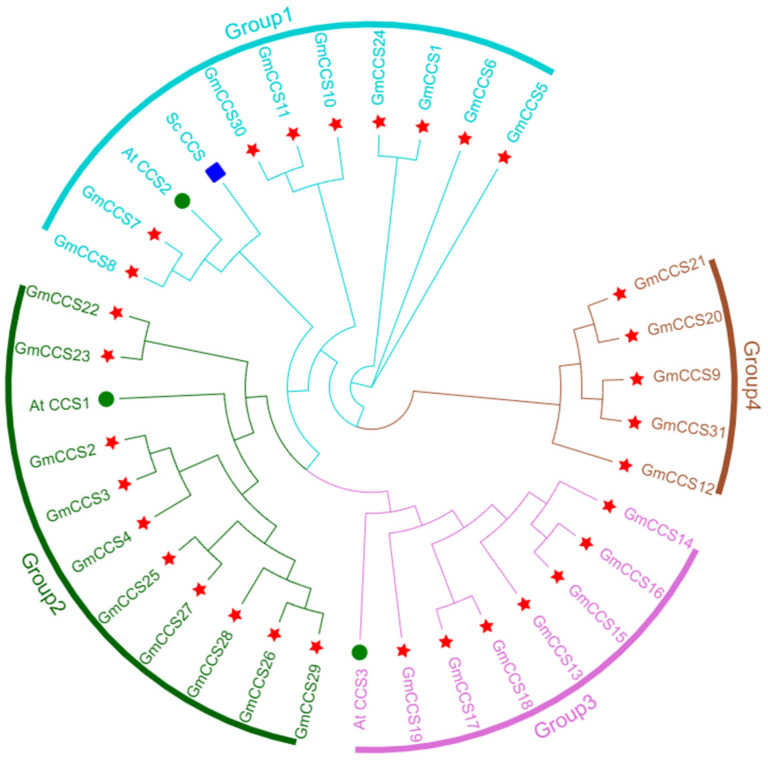
Phylogenetic tree of CCS proteins in *Glycine (G.) max*, *Arabidopsis thaliana* and *Saccharomyces cerevisiae*. Four subgroups were classified with different colors, and their names are marked in the corresponding positions. The circle, five-pointed star, and square text indicate *Arabidopsis thaliana*, *Glycine (G.) max* and *Saccharomyces cerevisiae*, respectively.

**Figure 3 ijms-24-05154-f003:**
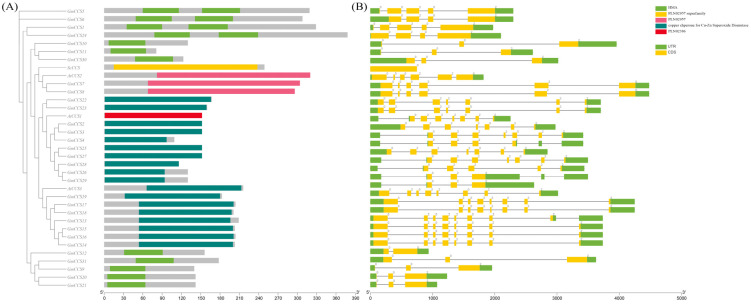
Gene structure, consvered domain analysis of GmCCS proteins from *Glycine (G.) max.* (**A**) The conserved domain in the GmCCS proteins. The green boxes indicate HMA domain (heavy-metal-associated), blue boxes respectively indicate Cu-zine (Zn) superoxide dismutase and the grey box indicate non-specific functional area. (**B**) The UTR, CDS and intron organization of GmCCSs. The green boxes represent UTRs, the yellow boxes represent CDS and thin black lines represent introns.

**Figure 4 ijms-24-05154-f004:**
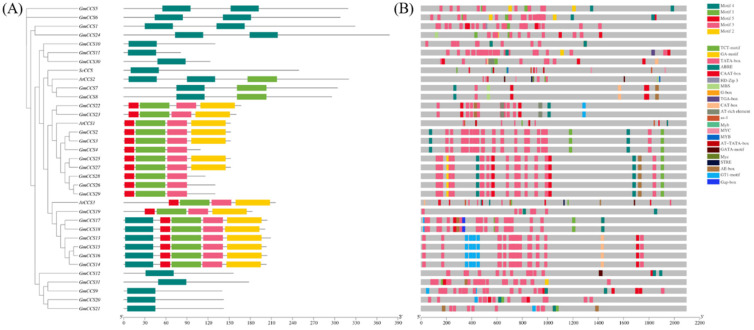
(**A**) The motifs in GmCCSs’ protein. Boxes 1–5 represent motifs 1–5, the grey line indicate the non-specific functional area. (**B**) Visualization of hormone-responsive elements in GmCCSs promoters by TBtools, including position, kind and quantity of elements.

**Figure 5 ijms-24-05154-f005:**
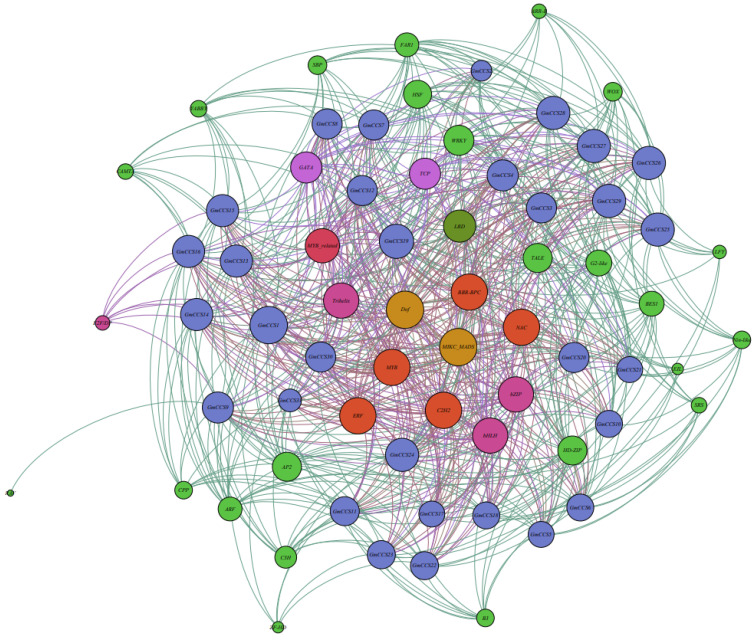
Interacting network among *GmCCS* family and their potential predicted TFs/protein. The blue cycle represents the GmCCSs gene, the circles with different colors represent different TFs, the lines indicate the relationship between different TFs.

**Figure 6 ijms-24-05154-f006:**
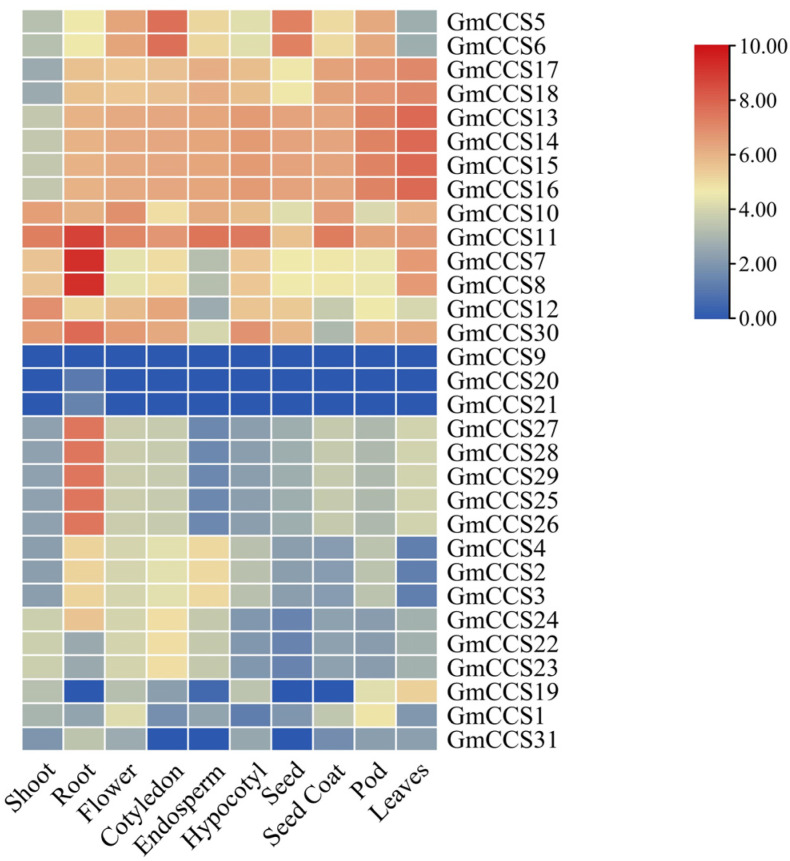
Heat map of expression profiles (in log2-based FPKM) of soybean *GmCCS* subgroup genes in different tissues (root, root hairs, stem, leaf, nodules, flower and seed). The gene names are on the left and the tissue names are on the top of the figure, and the expression abundance of each transcript is represented by the color bar: Red, higher expression; blue, lower expression.

**Figure 7 ijms-24-05154-f007:**
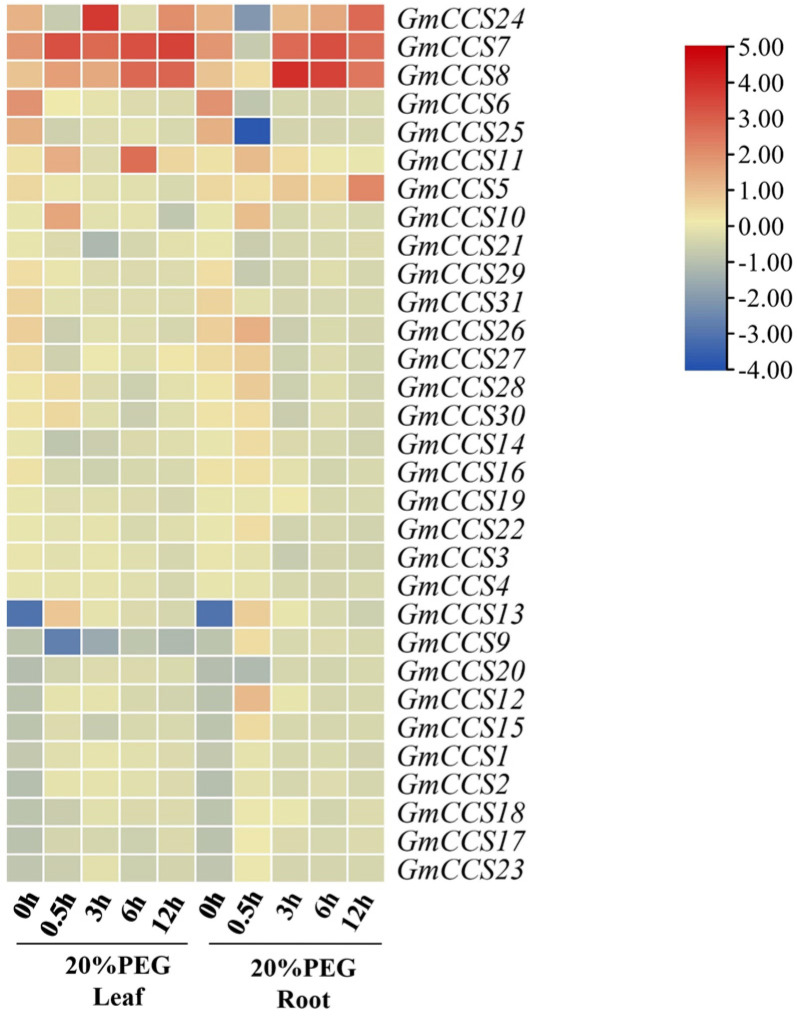
RT-qPCR analysis of *GmCCS* genes under drought stress. The heatmap was created based on the RT-qPCR results. The transcript abundance is represented by the color bar.

**Figure 8 ijms-24-05154-f008:**
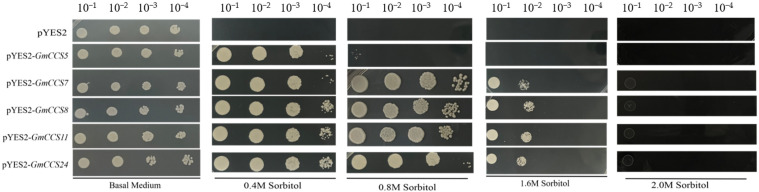
Drought tolerance test of candidate *GmCCS* genes in transformed yeast cells.

**Figure 9 ijms-24-05154-f009:**
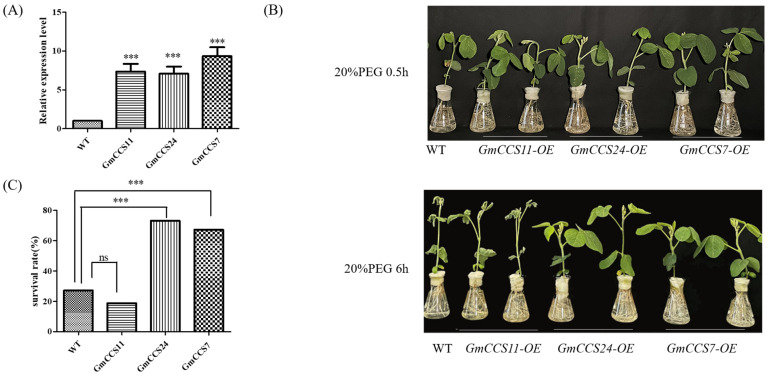
Soybean hairy roots expressing *GmCCS7*/*GmCCS24* showed improved drought stress tolerance. (**A**) RT-qPCR analysis of *GmCCS7*, *GmCCS11* and *GmCCS24* transcript levels in transgenic hairy roots. Transcript levels relative to *TUA5* were represented in each treatment; error bars indicate standard deviation (n = 3 plants). (**B**,**C**) Performance and survival rates of soybean seeding plants with transgenic hairy roots overexpressing *GmCCS7*/*GmCCS11*/*GmCCS24* under drought stress; error bars indicate standard deviation (n = 15 plants). (**D**–**G**) The antioxidant enzymes (POD, SOD and CAT) and the MDA content analysis under drought stress; error bars indicate standard deviation (n = 10 plants). The ns indicate none significant. One-tailed student’s *t*-test was used to generate the *p* values. Stars indicate significant difference (*** *p* < 0.001) compared to control plants (expressing empty vector).

## Data Availability

The data presented in this study are available in the Appendix A.

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
