# Peer review of "Genome-Wide Identification and Characterization of Copper Chaperone for Superoxide Dismutase (CCS) Gene Family in Response to Abiotic Stress in Soybean"

_ijms, 2023, doi:10.3390/ijms24065154_

Round 1

Reviewer 1 Report

This phrase  "Drought stress is one of the main environmental constraints that severely affects plant growth and development [1]" is quite repetitive respect to the following.

The introduction exposition could be a little more ordered. Grouping the information by blocks with a clear sense in its order of exposition as well as in their contents and interrelation betwen them. The brief reviewing of Cu as micronutrient, the cite about the CCS null mutant in Drosophila... Seems a little unlinked.

The labels from the collinear analysis (B) in Figure 1, are too small and difficult their reading. The orange bars corresponding to G.msax are no mentioned on legend.

There is not sufficiently explained why the authors select the three species employed in their comparative analysis, Glycine max, Arabidopsis thaliana and Saccharomyces cerevisiae. 

The philogenetic tree of CSS shows a clear unbalance in the number of 

On figure 6 the identification of legeng is wrong, the gene names are on the base and the tissue names on the right.

The labels from figure 7 are too small and its legend doesn't identify the differences betwen the A, B and C graphs.

The legends from all figures could be improved in clarity and utility.

The text would gain with a bigger structuration.

The bioinformatic tool Plant-mPLo 2.0, can't be found in the provided link  (http://www.csbio.sjtu.edu.cn)

Author Response

Point 1: This phrase "Drought stress is one of the main environmental constraints that severely affects plant growth and development [1]" is quite repetitive respect to the following.

Response 1: I rewrote this part of the description in line 34.

Point 2: The introduction exposition could be a little more ordered. Grouping the information by blocks with a clear sense in its order of exposition as well as in their contents and interrelation betwen them. The brief reviewing of Cu as micronutrient, the cite about the CCS null mutant in Drosophila... Seems a little unlinked.

Response 2: I rewrote this part of introduction.

Point 3: The labels from the collinear analysis (B) in Figure 1, are too small and difficult their reading. The orange bars corresponding to G.msax are no mentioned on legend.

Response 3: I have supplemented the description in the legend at line 111 and resized the (B) in Figure 1.

Point 4: There is not sufficiently explained why the authors select the three species employed in their comparative analysis, Glycine max, Arabidopsis thaliana and Saccharomyces cerevisiae.

Response 4: In our study, soybean CCS gene family members were main research object, the CCS was the first discovered and cloned in Saccharomyces cerevisiae, while yeast and Arabidopsis were also indispensable model test subjects for biological experiments.

Point 5: The philogenetic tree of CSS shows a clear unbalance in the number of

Response 5: The phylogenetic tree of CCS proteins including 31 GmCCS proteins and 3 AtCCS proteins and one ScCCS protein.

Point 6: On figure 6 the identification of legeng is wrong, the gene names are on the base and the tissue names on the right.

Response 6: In the initial mapping, in order to show the expression of 31 genes in different plant tissues, we selected the gene names are on the base and the tissue names on the right. Now the legend of figure 6 have been changed, the gene names were changed on the right and the tissue names on the base at line 194.

Point 7: The labels from figure 7 are too small and its legend doesn't identify the differences betwen the A, B and C graphs.

Response 7: I rearranged the presentation of Figure 7.

Point 8: The legends from all figures could be improved in clarity and utility.

Response 8:The legends of all figures were improved as far as possible.

Point 9: The text would gain with a bigger structuration.

Response 9: Thank you for your rigorous review and valuable comments, which prompted me to revise the manuscript carefully.

Point 10: The bioinformatic tool Plant-mPLo 2.0, can't be found in the provided link (http://www.csbio.sjtu.edu.cn)

Response 10: I have changed the link( http://www.csbio.sjtu.edu.cn/bioinf/plant-multi/ ) at line 369.

Reviewer 2 Report

This is an interesting study and the paper is well written and structured. I suggest just minor revision.

Author Response

Thank you for your encouragement and I will further improve the manuscript.

Reviewer 3 Report

- Line 41: remove superscript 1O2 and capital O instead of zero (0) for the superoxide radical.

-Line 55: Drosophila with italics.

-Gene names with italics. Check the whole text.

-Line 78: Arabidopsis italics.

-Line 100: Arabidopsis thaliana --> italics

-In discussion, there is a lack of previous corresponding references, please illustrate your findings demonstrating results from previous research works.

Author Response

Point 1:Line 41: remove superscript 1O2 and capital O instead of zero (0) for the superoxide radical.

Response 1: I have removed superscript of 1O2, the formula of superoxide radical was written using the letter O, which looked like zero(0).

Point 2:Line 55: Drosophila with italics.

Response 2:I have changed Drosophila with italics in line 55.

Point 3:Gene names with italics. Check the whole text.

Response 3: I carefully checked whether the descriptions of genes in the manuscript were in italics, and the descriptions of proteins were described in non-italics.

Point 4:Line 78: Arabidopsis italics.

Response 4: I have changed Arabidopsis with italics in line 78.

Point 5:Line 100: Arabidopsis thaliana --> italics

Response 5: I have changed Arabidopsis with italics in line 100.

Point 6:In discussion, there is a lack of previous corresponding references, please illustrate your findings demonstrating results from previous research works.

Response 6: I rewrote this part.

Round 2

Reviewer 1 Report

A final revision it would be desirable (spell, format,  etc ..) but in my opinion it is acceptable in its present form.

Reviewer 2 Report

Good luck with your research!

Reviewer 3 Report

After the changes the authors made, I believe that this article is ready to be accepted for publication.